# Effect of Bortezomib on Global Gene Expression in PC12-Derived Nerve Cells

**DOI:** 10.3390/ijms21030751

**Published:** 2020-01-23

**Authors:** Karolina Łuczkowska, Dorota Rogińska, Zofia Ulańczyk, Bogusław Machaliński

**Affiliations:** Department of General Pathology, Pomeranian Medical University, 70-111 Szczecin, Polanddoroginska@gmail.com (D.R.); z.litwinska@gmail.com (Z.U.)

**Keywords:** bortezomib, miRNA, microarray, neurotoxicity

## Abstract

Peripheral neuropathy is one of the main side-effects of novel therapeutics used in oncohematological diseases, but the molecular basis underlying its development and progression as well as neurotoxicity mechanisms induced by the use of these therapeutics are still not fully elucidated. The aim of this study was to demonstrate the effect of bortezomib on global gene and miRNA expression on PC12-derived nerve cells. Microarray analysis showed that expression of 1383 genes was downregulated at least two fold and 671 genes were upregulated at least two fold in PC12-derived nerve cells treated with bortezomib compared to untreated/control cells. Analysis of functional annotations mainly identified downregulated processes (e.g., regulation of cell cycle, DNA replication and repair, regulation of cell migration, neuron projection morphogenesis and neurotransmitter secretion). The result of miRNA expression analysis demonstrated only 11 significantly downregulated miRNAs (at least two fold) in bortezomib-treated PC12-derived nerve cells vs. control cells. MiRNAs regulate gene expression, therefore we decided to conduct an analysis comparing the outcomes of miRNA microarray expression data to the obtained mRNA data. The most interesting miRNA–target gene correlation is downregulated expression of miR-130a-3p and miR-152-3p and as a result of this downregulation the expression of the *Gadd45* increased. This gene is a member of a group of genes, the transcript expression of which is enhanced after stressful growth arrest conditions and treatment with DNA-damaging agents like drugs or mutagens.

## 1. Introduction

Multiple myeloma (MM), which makes up around 10% of all haematological malignancies, is a malignant plasma cell disorder [1]. A major step forward in MM treatment has been the introduction of so-called novel agents, including the immunomodulatory drug (IMID) thalidomide, the subsequent development of its potent analogue lenalidomide, and the first-in-class proteasome inhibitor bortezomib (BZT). These agents serve a variety of anti-MM functions, including inhibition of MM cell proliferation and angiogenesis, apoptosis induction, and regulation of interactions between MM cells and bone marrow stromal cells [2]. Proteasome inhibitors are drugs that inhibit the ubiquitin-proteasome pathway and affect multiple signalling cascades in the cells. Bortezomib contains boron in its chemical structure and reversibly inhibits the chymotrypsin-like activity of the 26S proteasome degrading ubiquitinated protein, thus enabling the cell to maintain the balance of regulatory proteins involved in the cell cycle and division. Therefore, inhibition of proteasome activity by bortezomib leads to loss of cell cycle control and cell death [3]. BZT also decreases adhesion of myeloma plasma cells to stromal cells and disrupts secretion of specific cytokines in the bone marrow microenvironment such as vascular endothelial growth factor (VEGF), insulin-like growth factor 1 (IGF-1), interleukin-6 (IL-6), and tumor necrosis factor-α (TNF-α) [4]. The additional effect of bortezomib is based on the inhibition of DNA repair and angiogenesis, impairment of osteoclast activity [5] and disruption of intracellular calcium metabolism. This can lead to mitochondrial calcium inflow and apoptosis induced by caspase activation [6]. In addition, mitochondrial damage leads to oxidative stress through activation of ion channel TRPA1 in nociceptors.

From the early days of anti-MM agents’ clinical use, it has become apparent that peripheral neuropathy (PN) is one of their main non-haematological dose-limiting side-effects [7]. Bortezomib-induced peripheral neuropathy (BiPN) occurs in up to 40% of patients [8] and is related to the dose, schedule, and mode of administration [9]; it is usually reversible. Neurotoxicity can significantly affect the quality of a patient’s life, and may require dose-reduction, delay, or even premature termination of treatment. One of the hypothesised developments of neuropathy is the specific neurotoxic effect of bortezomib. BZT affects the transient release of intracellular calcium supply, leading to mitochondrial calcium influx and caspase-induced apoptosis. It has also been suggested that bortezomib exposure is closely associated with post-transcriptional gene silencing mediated regulation by non-coding RNAs, mainly microRNAs (miRNAs, miRs) [10,11]. MiRNAs are 20–22 nucleotide long molecules, which regulate gene expression and are involved in practically all biological processes, including apoptosis, cell proliferation, as well as disease-associated mechanisms [12]. According to the available literature, changes in miRNA expression are characteristic as a response to drug use in a number of diseases. [13]. However, the wide panel of mRNAs/miRNAs has not yet been tested in bortezomib-treated PC-12-derived nerve cells.

To better understand the underlying molecular mechanisms of bortezomib-induced neurotoxicity, in this study we analyzed the global expression of mRNA and miRNA in bortezomib-treated PC12-derived nerve cells and examined the changed biological pathways and miRNA–mRNA target interactions.

## 2. Results

### 2.1. Gene Expression Profile in PC12-derived Nerve Cells

Microarray analysis showed that expression of 1383 genes was downregulated at least two fold (fold -42.49 to -2) and 671 genes were upregulated at least two fold (fold 78.3 to 2) in PC12-derived nerve cells treated with bortezomib compared to untreated/control cells (Figure 1). The detailed fold values of most 15 up- or downregulated genes are presented in Table 1 and Table 2, respectively. Next, up/downregulated genes were classified according to the Gene Ontology (GO) classification of biological processes (Figure 2) and the KEGG Database of pathways (Figure 3). Analysis of functional annotations mainly identified downregulated processes (e.g., regulation of cell cycle, DNA replication and repair, regulation of cell migration, neuron projection morphogenesis, and neurotransmitter secretion). In addition, a detailed analysis of two pathways was performed (Figure 4: Axon guidance; Figure 5: Cell cycle).

Since treatment-induced neurotoxicity can be caused by a number of factors, in our analysis of microarray data we focused on several important processes that affect nervous system homeostasis. First, we focused on dysregulated processes essential for neural tissue function, such as neuron projection morphogenesis and neurotransmitter secretion, and genes involved in these processes that showed significant fold change in our study (Figure 6). Figure 7 presents more generalized processes that were also significantly dysregulated in our study (DNA repair, regulation of cell cycle, cell division), and genes involved in these processes that showed expression changes in our study.

#### Gene Set Enrichment Analysis

The Gene Set Enrichment Analysis (GSEA) method focuses on gene sets, that is, groups of genes that participate in a common biological function, chromosomal location, or regulation. Therefore, to more precisely investigate the biological influence of bortezomib on PC12-derived nerve cells, we carried out GSEA to identify altered pathways in bortezomib-treated PC12-derived nerve cells vs. control cells. According to the gene set enrichment analysis, 10 gene sets with a positive correlation were detected in bortezomib-treated cells and 10 gene sets with a negative correlation. Cell cycle checkpoints, DNA repair, and neuronal system-related genes were significantly downregulated in PC12-derived nerve cells treated with bortezomib for 24 h (false discovery rate (FDR) = 0.61%) (Figure 8).

### 2.2. Validation of Dysregulated mRNAs

We chose significantly dysregulated genes (*Ascl1, Top2a, Slc7a11, Ddit3*) to perform validation by qRT-PCR. In all selected genes, the results of qRT-PCR confirmed the result of expression of selected mRNAs obtained by global gene the expression analysis (Figure 9).

### 2.3. miRNAs Expression Profile in Neural Cells

The result of miRNA expression analysis demonstrated only 11 significantly downregulated miRNAs (at least two fold) in bortezomib-treated PC12-derived nerve vs. control cells (Figure 10).

MiRNAs regulate gene expression, therefore we decided to conduct an analysis comparing the outcomes of miRNA microarray expression data to the obtained mRNA data. Only this method shows the real impact of miRNAs on their target genes. Figure 11 shows changes in the expression of 9 miRNAs; target genes and changes in their expression in bortezomib-treated PC12-derived nerve compared to control cells. The identified miRNAs regulate various processes: Cell cycle, apoptotic process, regulation of cell growth.

### 2.4. Validation of Dysregulated miRNAs

We chose significantly dysregulated miRNAs (miRNA-21-5p; miRNA-322-5p; miRNA-532-5p) to perform validation by qRT-PCR. In all selected miRNAs, the results of qRT-PCR confirmed the result of expression of selected miRNAs obtained by global gene expression analysis (Figure 12).

## 3. Discussion

Since the incidence of chemotherapy-induced peripheral neuropathy in patients with MM treated with novel agents is still significant, the issue of the adverse effects of these drugs remains the important clinical problem. Suggested mechanisms underlying drug-induced PN comprise direct damage to dorsal root ganglion (DRG), mitochondrial and endoplasmic reticulum damage, deregulation of Ca^2+^ homeostasis, inhibition of neurotrophins’ transcription, microtubule stabilization, and autoimmune or inflammatory factors [14]. In this study, we demonstrated for the first time the global effect of bortezomib on expression of genes and miRNAs associated with neurogenesis, cell cycle, cell division, and DNA repair in PC12-derived nerve cells.

The introduction of bortezomib has been a significant breakthrough in the therapy of MM. BZT has a lot of different anti-myeloma effects including disruption of cell cycle, induction of apoptosis, alteration of the bone marrow microenvironment, and inhibition of nuclear factor kappa B (NFκB) [15]. The additional effect of bortezomib is based on the inhibition of DNA repair and angiogenesis, impairment of osteoclast activity [5], and disruption of intracellular calcium metabolism [16]. Recent studies have shown that bortezomib causes acute cytoskeletal damage, which is relevant in neural cell integrity. In addition, it has been demonstrated that bortezomib causes excessive protein carbonylation and destabilization of actin filaments [17]. Our current study provided interesting results that indicate a significant dysregulation of genes in neuronal cells, which may be characteristic for bortezomib activity (Figure 2, Figure 3 and Figure 8).

The DNA damage, for example altered bases and single- and double-strand breaks, is an important neurotoxic factor suggested in the aging and pathogenesis of many neurological disorders, including amyotrophic lateral sclerosis (ALS), Parkinson’s disease, and Alzheimer’s disease. DNA double-strand break (DSB) repair occurs in two ways, via: a) Homologous recombination, which uses a sister chromatid, and b) non-homologous end joining (NHEJ). DSB in the DNA cannot be completely repaired by the aforementioned repair ways in mature neurons because they are unable to actively divide. Therefore, it is considered that neurons accumulate DNA damage, which could potentially be the basis for the pathogenesis of neurodegenerative disorders [18]. DNA of peripheral nervous system (PNS) is more exposed to endo- and exogenous factors compared to the central nervous system (CNS) because the blood brain barrier (BBB) provides good protection [19]. One of the most downregulated genes in our study after bortezomib-treatment is topoisomerase (*Top2a*) (Table 1), which encodes an enzyme necessary for normal development. Topoisomerase relaxes coiled DNA to eliminate helical constraints, which may impede DNA replication and transcription and consequently inhibit cell growth [20]. Mutations in *TDP1* play an important role in the development of neurodegenerative diseases such as ataxia telangiectasia (A-T) and spinocerebellar ataxia with axonal neuropathy (SCAN-1) [21]. We also observed reduced expression of another topoisomerase, topoisomerase (DNA) II binding protein 1 (*Topbp1*) (fold change = −7.37), which prevents DNA double-strand breaks. Lee et al. confirmed that TopBP1 is crucial for preventing replication-associated DNA strand breaks, but it is not necessary for replication [22]. In addition, they concluded that TopBP1 is essential for providing genome integrity in the early progenitors that direct neurogenesis. The basis for single-strand breaks (SSBs) repair is to generate DNA ends that enable ligation. DNA ligases play an important role in this process because they allow production of a phosphodiester bond to join broken DNA strands together. They play an important role in DNA replication, nuclear DNA metabolism, nucleotide excision repair (NER), single-strand break repair, homology-mediated DSB repair, and base excision repair (BER) [23]. In our study we observed decreased expression of *Lig1* gene (fold = −3.605), which contribution to the development of neuropathy has been described [24]. The next gene with decreased expression (fold = −2.052) involved in the DNA repair process is *Ercc6*, responsible for the peripheral repair of transcribed sequences. It participates in the regulation of transcription and DNA repair, and ensures the stability of chromosomes. Troelstra et al. have demonstrated in a patient with Cockayne’s syndrome (CS) that *ERCC6* is not crucial for cell viability but is specific for preferential repair of transcribed sequences [25].

In clinical practice (especially cancer treatment), drugs, such as cisplatin, bortezomib, or methotrexate, that affect apoptosis, DNA repair, or the cell cycle are used. Despite the effectiveness of these drugs in therapy, they also affect healthy tissues, causing side effects. Cisplatin is an anticancer drug that causes intra- and inter-strand crosslinks in the DNA. Cisplatin has a neurotoxic effect on the peripheral nervous system which is directly related to DNA damage. Dzagnidze et al. observed in mice with dysfunctional deficient for NER that chronic exposure to cisplatin caused accelerated accumulation of unrepaired intrastrand cross-links in neuronal cells, and was significantly correlated to the degree of sensory impairment as measured by electroneurography [26].

We also observed low expression in *Kif5A* gene (fold change = −3.32) in our study. *KIF5A* plays a significant role in the transport of organelles in the nerve cells and transfer of mitochondria from the neural soma to the axon terminal [27]. Nam et al. demonstrated that mutations reducing *KIF5A* expression are important in the development of hereditary spastic paraplegia (HSP) and axonal Charcot–Marie–Tooth peripheral neuropathy type 2 (CMT2). Reduced expression of this gene leads to axon degeneration in the peripheral nervous system [28]. The lipoprotein lipase (*Lpl*) gene had the most reduced expression in our study (fold change = −42.49). LPL supplies the lipids for myelin sheath by hydrolysing the triglyceride rich lipoproteins. Recent research by Rachana et al. has shown that LPL is an important factor in the regeneration of myelin sheath during development of diabetic neuropathy [29].

Our next interesting observation is the inhibition of mitogen-activated protein kinase/extracellular signal-regulated kinase (MAPK/ERK) signaling (Figure 4). Pathways such as extracellular signal regulated kinase1/2 (*ERK1/2*), phoshpatidyloinositol-3-kinase (*PI3K*), or extracellular signal regulated kinase5 (*ERK5*) are responsible for neuronal survival signaling [30] and mediating the transmission of signals from the synapse to cytoplasmic and nuclear effectors [31]. Davis et al. presented on a rat model that MAPK/ERK signaling is crucial for neuronal plasticity and long-term memory formation [32].

MicroRNAs (miRNA) are small, conserved non-coding RNA molecules involved in the regulation of gene expression. MiRNAs are also markers of disease diagnosis and prognosis [33]. In our study, we observed a small number of miRNA–target gene correlations. Thus, we hypothesize that the observed significant changes in gene expression after bortezomib exposure could probably be attributed to other mechanisms. Nonetheless, we have seen a very significant connection. Namely, expression of rno-miR-130a-3p and rno-miR-152-3p was downregulated and as a result the expression of the *Gadd45* increased (fold change = 5.91). This gene is a member of a group of genes, the transcript expression of which is enhanced after stressful growth arrest conditions and treatment with DNA-damaging agents like drugs or mutagens [34].

### Potential Study Limitations

Our study showed significant results; however, its weakness is the use of one dose of bortezomib at one time point. In clinical practice, repeated administration of bortezomib is used; therefore, in order to extensively assess the role of bortezomib in the development of drug-induced peripheral neuropathy in the treated patient, it would be advantageous to also use multiple doses in in vitro studies. However, our study was not designed nor extensive enough to unequivocally show molecular basis of BiPN. Our study rather presents miRNA and gene expression changes after single bortezomib exposure in PC12-derived nerve cells. While the nature of this study was rather preliminary, we believe it provides a solid basis for future studies that could fully explain the effect of bortezomib on the development of neuropathy.

## 4. Materials and Methods

### 4.1. Cell Culture and Incubation

PC12 pheochromocytoma cells (rat, ATCC, Manassas, VA, USA) were used in this study. The selected cell line is often used as a reliable model of induced peripheral neuropathy and neurotoxicity [35,36,37,38,39]. The cells respond reversibly to nerve growth factor (NGF) by induction of the neuronal phenotype when plated on collagen IV coated culture flasks. PC12 cells were incubated for growth in Corning^®^ T-75 flasks (Corning, MO, USA) in RPMI-1640 (ATCC, Manassas, VO, USA) medium modified to contain 2 mM L-glutamine, 10 mM HEPES, 1 mM sodium pyruvate, 4500 mg/L glucose, and 1500 mg/L sodium bicarbonate, streptomycin (100 µg/mL), penicillin (100 U/mL) with 5% fetal bovine serum (FBS) (Sigma Aldrich, St. Louis, MO, USA) and 10% heat-inactivated horse serum (HS) (Sigma Aldrich, St. Louis, MO, USA) at 37 °C in saturated humidity atmosphere containing 5% CO_2_. The medium was changed every 2–3 days. PC12 cell differentiation into neurons was conducted according to the protocol described by Popova et al. [39]. PC12 cells were incubated at a density of 2x10^6^ cells/well in 6-well culture plates on collagen IV coated (Corning, MO, USA) in RPMI-1640 medium supplemented with 2 mM L-glutamine, 10 mM HEPES, 1 mM sodium pyruvate, 4500 mg/L glucose, and 1500 mg/L sodium bicarbonate, streptomycin (100 µg/mL), penicillin (100 U/mL), 1% HS, and 100 ng/mL NGF (Sigma Aldrich, St. Louis, MO, USA). Every second day, half of the medium/well was changed. After 10 days, differentiated PC12 cells into neurons were received (Figure 13). Next, neurons were not treated (control) or treated with bortezomib 50 nM/L (Cell Signalling Technology, Danvers, MA, USA) and collected after 24 h of incubation. The cells were used to isolate mRNA and miRNA.

### 4.2. RNA

RNA was isolated from three separate cell incubations for each group. Total RNA was isolated from PC12-derived nerve cells (1.5 × 10^6^) using the mirVana™ miRNA Isolation Kit (Thermo Fisher, Waltham, MA, USA) following the manufacturer’s instructions. The used kit allows RNA and miRNA isolation simultaneously. Quality and concentration of the received RNA were evaluated using Epoch spectrophotometer (Biotek, Winooski, VT, USA).

### 4.3. Affymetrix GeneChip Microarray and Data Analysis

RNA was isolated from three separate cell incubations. Next, the isolated total RNA samples were pooled to perform microarray analysis. mRNA was used to produce a complementary single-stranded DNA (cDNA) using an Ambion WT Expression Kit (Thermo Fisher Scientific, Waltham, MA, USA) and labelling with the GeneChip WT Terminal Labeling Kit (Affymetrix, Santa Clara, CA). After this step of the procedure, hybridization was performed on an Affymetrix Rat Gene 2.1 ST Array Strip. Hybridization, washing, and scanning were performed with an Affymetrix GeneAtlas™ System (Affymetrix, Santa Clara, CA, USA). The generated raw data were imported into the BioConductor software based on the statistical R programming language. The Robust Multiarray Averaging (RMA) algorithm implemented in the “affy” package of BioConductor was used for background normalization, correction, and summation of raw data. The BioConductor “oligo” package was used to obtain biological annotation.

### 4.4. Affymetrix GeneChip miRNA Microarray and Data Analysis

Total RNA-enrichment in miRNAs was isolated from three separate cell incubations. Next, the isolated total RNA samples were pooled to perform microarray analysis. The first step of the procedure was a poly (A) tailing reaction and ligation of the biotinylated signal molecule to the target RNA. Next, the sample was hybridized onto an Affymetrix miRNA 4.1 Array Strip (Affymetrix, Santa Clara, CA, USA). Finally, streptavidin-PE was added and array was prepared for scanning with the Affymetrix GeneAtlas system (Affymetrix, Santa Clara, CA, USA). Microarray data analysis was performed using BioConductor [40]. Normalized data were combined with the “pd.mirna.4.1” description file, including: Types, names, and sequences of miRNAs. The linear models for microarray data implemented in the “limma” library was used to determine differential expression [41].

A database miRTarBase was used to download the list of experimentally validated miRNA target genes. Only targets genes for changed expressed miRNA in the study were subtracted from the whole rat miRNA–target dataset. The target gene list was subjected to functional annotation and clusterization using the DAVID (Database for Annotation, Visualization, and Integrated Discovery) [42]. Target gene symbols of differentially expressed miRNA were uploaded to DAVID by the “RDAVIDWebService” BioConductor library [43], where targets were assigned to suitable Gene Ontology (GO) terms.

The miRNA–target gene interaction network was performed using “networkD3” library. Given that miRNAs mainly function as the repressor of the transcription of target genes, only target genes that negatively correlated with miRNAs were presented. If the expression of miRNAs was upregulated, the expression of its target genes should be downregulated and inversely. With such an assumption, interaction network was performed for all changed miRNA expressions, simultaneously considering only target genes with significantly altered expression.

### 4.5. DAVID

DAVID Bioinformatics Resources (Database for Annotation, Visualization, and Integrated Discovery) at http://david.abcc.ncifcrf.gov is a functional annotation and enrichment analysis. Functional annotation charts generated by DAVID with overrepresented gene annotations are shown as bubble plots from the BACA BioConductor package (https://cran.r-project.org/web/packages/BACA/BACA.pdf) [42,43].

### 4.6. Validation of Data Obtained from Microarrays

To validate the expression of genes and miRNA the qRT-PCR method was used. Only genes and miRNAs with significantly changed expression were selected for validation (fold >2). Reaction was performed on a Bio-Rad CFX96 Real-Time PCR Detection System (Bio-Rad Inc., Hercules, CA, USA). All primers for gene validation were designed by BLAST PRIMER and for miRNAs by miRPrimer, and purchased from the Laboratory of DNA Sequencing and Oligonucleotide Synthesis, Institute of Biochemistry and Biophysics, Polish Academy of Sciences, Warsaw, Poland. The qRT-PCR program consisted of: 10-min initial denaturation at 95 °C, denaturation at 95 °C for 15 s, annealing at 56 °C for 5 s, and extension at 72 °C for 10 s. The relative gene expression was quantified using the comparative Ct method (2ΔCt, where ΔCt = [Ct of target genes] − [Ct of endogenous control gene]). All products were characterized by high specificity, which was checked by determining melting points (0.1 °C/s transition rate).

#### 4.6.1. Validation of Gene Expression

The validation method using the qRT-PCR reaction consists of two main steps: (1) Reverse transcription of mRNA (0.1 µg) using the First Strand cDNA Synthesis Kit (Thermo Fisher Scientific, Waltham, MA, USA) and (2) main qRT-PCR reaction. The reaction mixture per sample consisted of: 7.5 μL of SYBR Green PCR Master Mix (Bio-Rad, Hercules, CA, USA); 1.5 μL cDNA template; 1.8 μL specific primers (0.9 μL reverse primer and 0.9 μL forward primer), and 4.2 μL Nuclease-Free Water. *Gapdh* gene was used as an endogenous control gene.

#### 4.6.2. Validation of Selected miRNA Expression

miRNA reverse-transcription was performed using qScript microRNA cDNA Synthesis Kit (Quanta Biosciences, Beverly, MA, USA) following the manufacturer’s instructions. The mixture for the qRT-PCR reaction contained: 1 μL microRNA cDNA; 5 μL PerfeCTa SYBR Green SuperMix; 0.2 μL microRNA specific primer; 0.2 μL PerfeCTa Universal PCR Primer; 4.6 μL Nuclease-Free Water. miR93 was used as an endogenous control gene.

### 4.7. Gene Set Enrichment Analysis (GSEA)

GSEA is a computational method focused on gene sets, that is, groups of genes that participate in common biological functions, chromosomal location, or regulation. The method uses for calculations the Kolmogorov–Smirnov (K-S) statistical test to identify significantly enriched or depleted groups of genes [44]. GSEA analysis was conducted using the GSEA Java Desktop Application from the Broad Institute (http://software.broadinstitute.org/gsea/index.jsp). Normalized data from all genes were transformed into a suitable format and imported into the application (GSEA Java Desktop Application, http://software.broadinstitute.org/gsea/index.jsp). Then, the Molecular Signatures Database (MsigDB) was selected to create a predefined gene set database [45]. The signal-to-noise ratio with 1000 permutations allowed classified genes into the selected set according to the difference in their expression level. Calculation of the enrichment score (ES) for each selected gene set was performed using a sum statistic [45]. Enrichment scores were normalized by their gene set size, and false-positive findings were corrected by FDR. Significant gene sets were considered to be those with an adjusted nominal *p* value < 0.01 and FRD *q* value < 25%.

### 4.8. Statistical Methods

Arithmetic means and standard deviations were calculated using MS Excel. Parameter comparisons between the two groups were performed using the unpaired Student’s t-test, and a *p*-value of <0.05 was considered statistically significant.

## 5. Conclusions

The study shows the effect of single exposure to bortezomib on gene and miRNA expression in PC12-derived nerve cells. We have shown that bortezomib negatively affects the cell cycle, DNA repair, nerve processes and cell division. Disorders of these processes could have a negative influence on peripheral nervous system homeostasis and may be associated with neurotoxicity of bortezomib. Further studies on the development of drug-induced peripheral neuropathy are required to provide in-depth explanation of bortezomib action in the neural system.

## Figures and Tables

**Figure 1 ijms-21-00751-f001:**
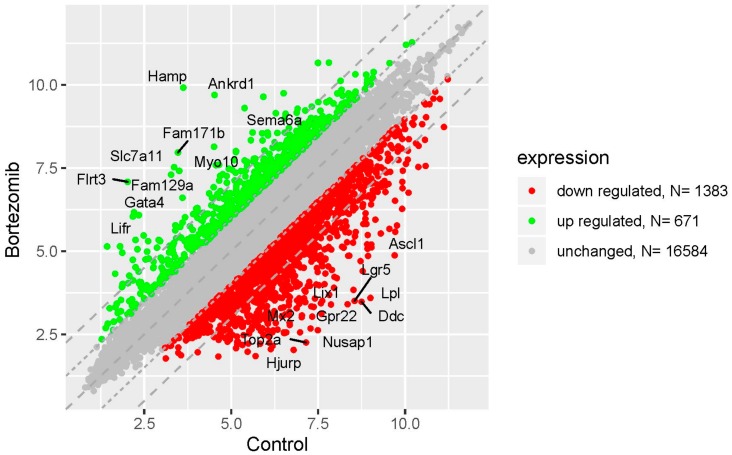
The scatter plot of global gene expression in bortezomib-treated PC12-derived nerve cells compared to the control cells. Genes are represented by dots (red colour: Downregulation; green colour: Upregulation). The graph shows the genes with at least twofold change and *p* < 0.05. The graph also contains the symbol of the genes with the largest change in expression.

**Figure 2 ijms-21-00751-f002:**
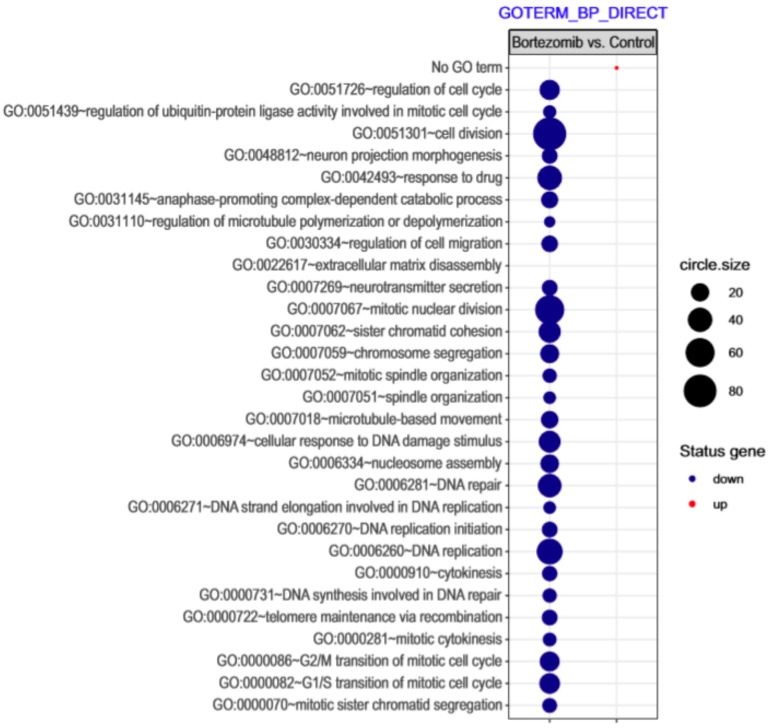
The bubble plot with changed biological processes assigned according to Gene Ontology (GO) classification in bortezomib-treated PC12-derived nerve cells compared to the control cells. Genes assigned to individual processes fulfilling the criteria: Adjusted *p* < 0.05, method = Benjamini, and minimum number of genes per group = 5, are presented. The bubble size indicates the number of genes represented in the corresponding annotation.

**Figure 3 ijms-21-00751-f003:**
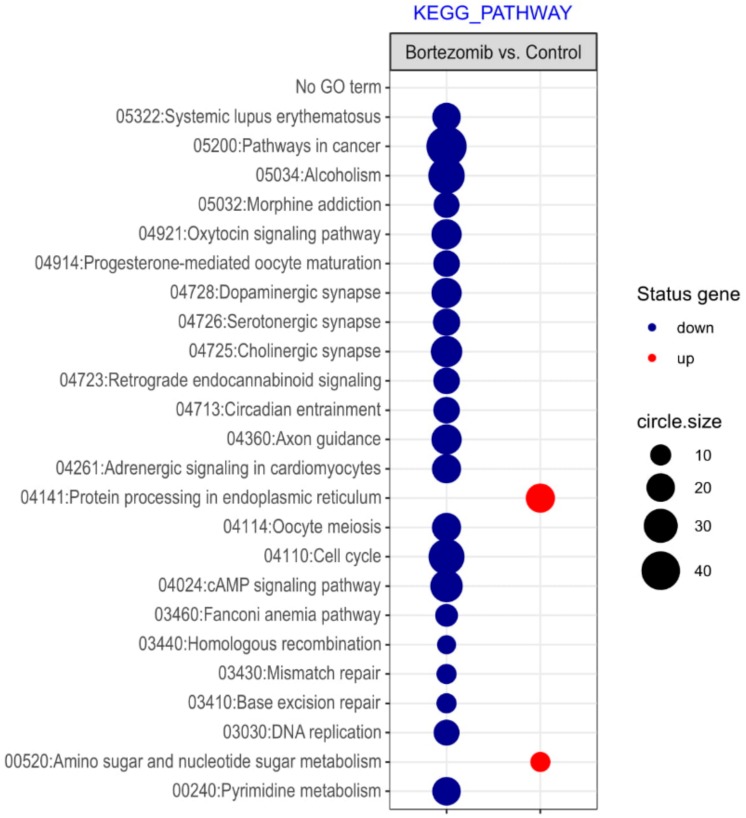
The bubble plot with changed pathways assigned according to the KEGG Pathway Database in bortezomib-treated PC12-derived nerve cells compared to the control cells. Genes assigned to individual processes fulfilling the criteria: Adjusted *p* < 0.05, method = Benjamini, and minimum number of genes per group = 5, are presented. The bubble size indicates the number of genes represented in the corresponding annotation.

**Figure 4 ijms-21-00751-f004:**
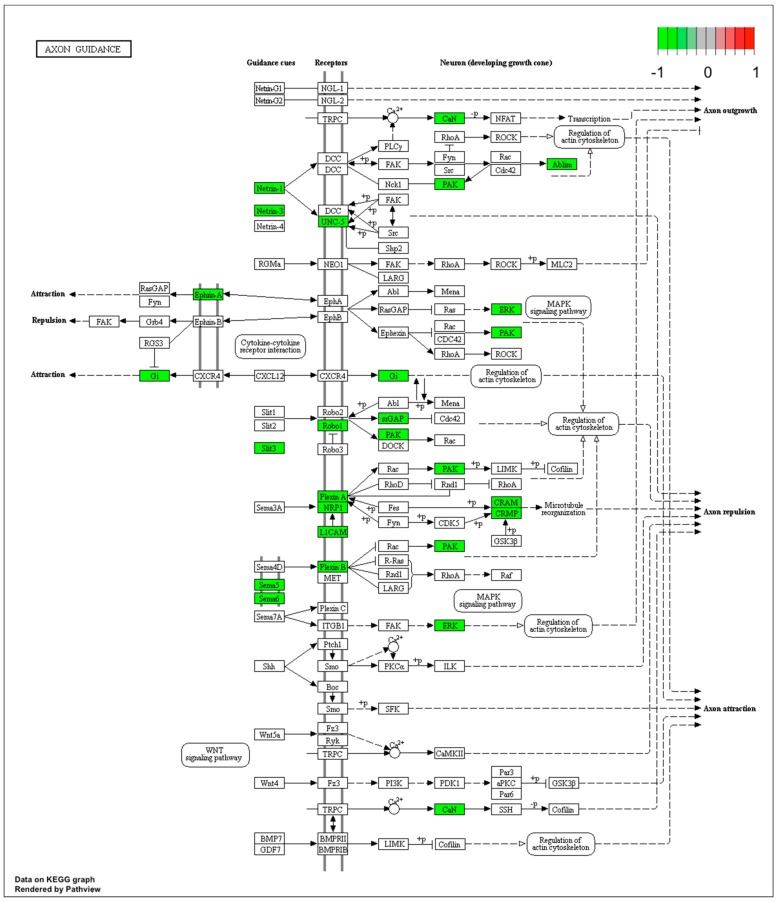
Graph of genes and processes involved in the pathway 04360.Axon guidance. The scheme shows genes and relationships between them. Additionally, the diagram indicates the genes up- or downregulated in our analysis. Solid arrows indicate a direct effect on specific gene expression while dotted arrows indicate gene involvement in a process or pathway. This allows an in-depth understanding of the effects of specific gene dysregulation on the selected process.

**Figure 5 ijms-21-00751-f005:**
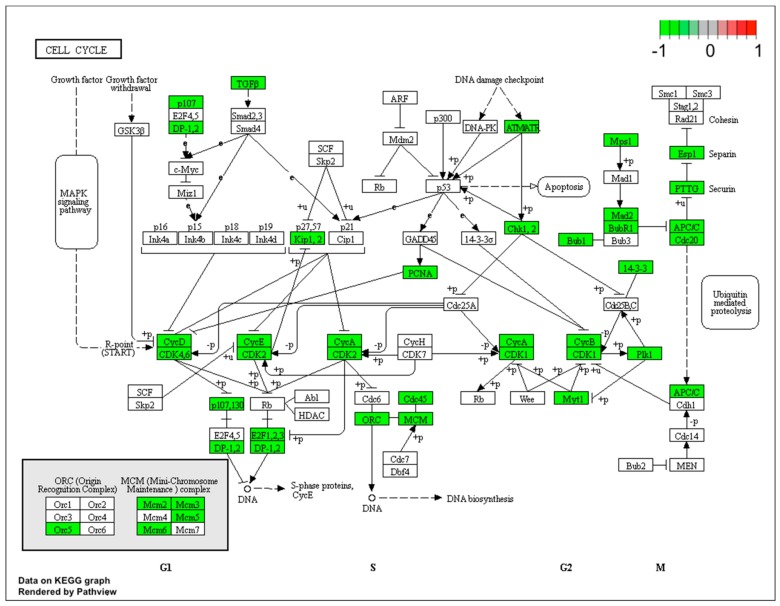
Graph of genes and processes involved in the pathway 04110: Cell cycle. The scheme shows genes and relationships between them. Additionally, the diagram indicates the genes up- or downregulated in our analysis. Solid arrows indicate a direct effect on specific gene expression while dotted arrows indicate gene involvement in a process or pathway. This allows an in-depth understanding of the effects of specific gene dysregulation on the selected process.

**Figure 6 ijms-21-00751-f006:**
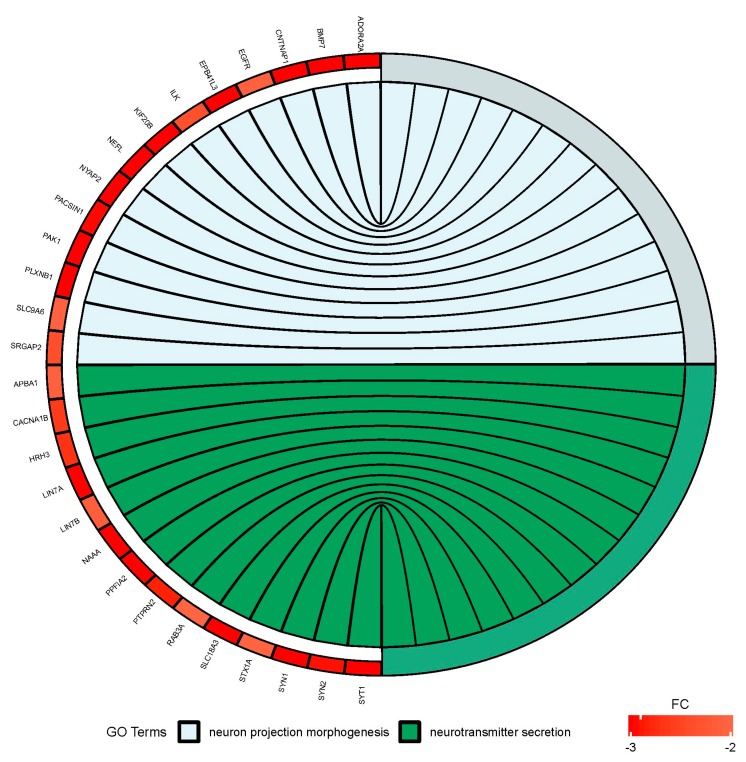
Circus plot shows the dysregulated processes (neuron projection morphogenesis and neurotransmitter secretion) and their associated genes after bortezomib-treatment in PC12-derived nerve cells. The level of expression for each gene is marked using red color (fold change −3 to −2).

**Figure 7 ijms-21-00751-f007:**
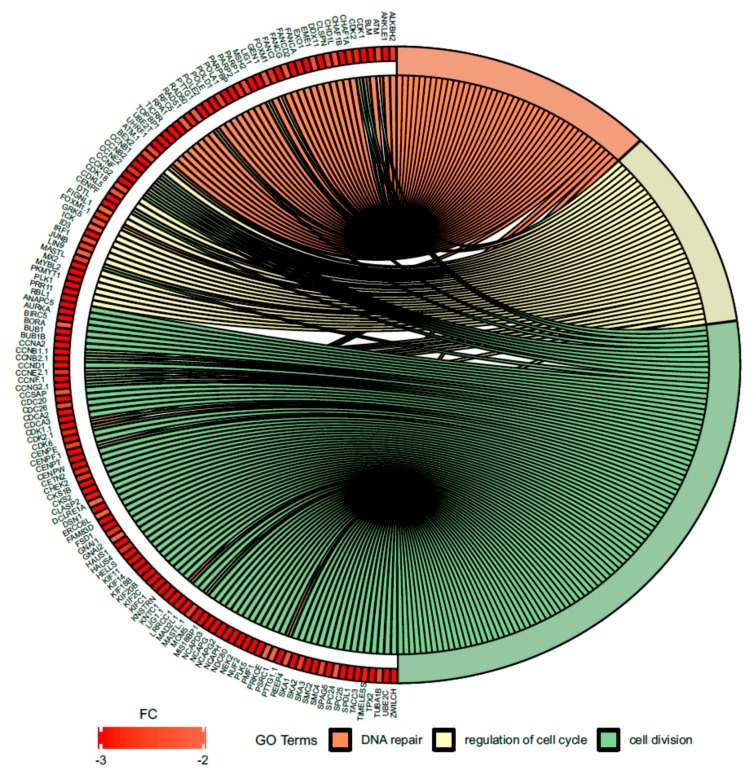
Circus plot shows the dysregulated processes (DNA repair, regulation of cell cycle, cell division) and their associated genes after bortezomib-treatment in PC12-derived nerve cells. The level of expression for each gene is marked using red color (fold change −3 to −2). Ribbons connecting areas of the Circus plots also indicate shared genes between groups.

**Figure 8 ijms-21-00751-f008:**
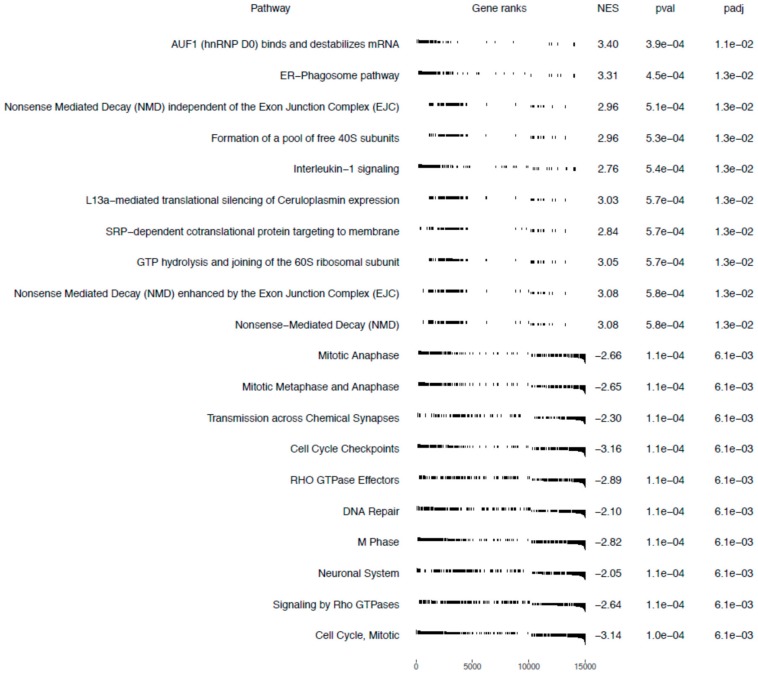
Gene sets enriched in bortezomib-treatment PC12-derived nerve cells. Gene sets are ranked according to the normalized enrichment score (NES).

**Figure 9 ijms-21-00751-f009:**
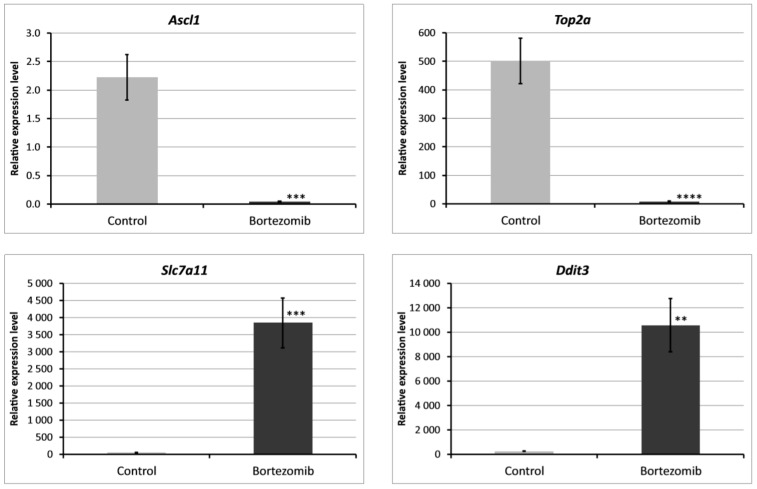
Real-time quantitation of selected genes *Ascl1, Top2a, Slc7a11, Ddit3* in PC12-derived nerve cells after bortezomib-treatment, and in the control cells. Data are presented as the mean ± SD (n = 3). ** *p* < 0.002; *** *p* < 0.001; **** *p* < 0.0009 compared with the control cells.

**Figure 10 ijms-21-00751-f010:**
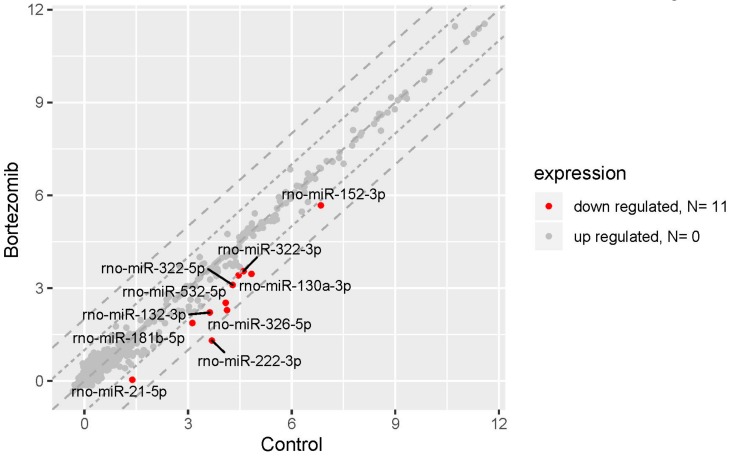
The scatter plot of global miRNAs expression in bortezomib-treated PC12-derived nerve cells compared to the control cells. Downregulated miRNAs are represented by red dots. The graph shows the miRNA with at least twofold change and p<0.05. The graph also contains the symbol of the miRNA with the largest change in expression.

**Figure 11 ijms-21-00751-f011:**
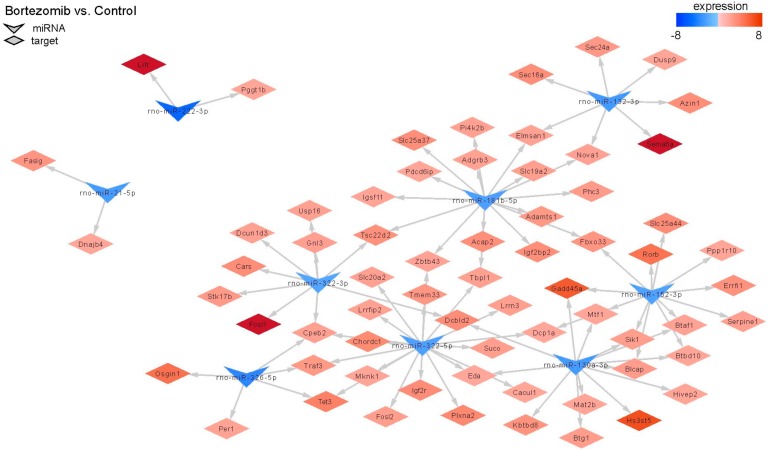
The diagram shows significantly downregulated miRNAs (at least fold −2). Target genes are assigned to each miRNA with a marked change in expression (at least fold −2) (red indicates upregulation and blue indicates downregulation).

**Figure 12 ijms-21-00751-f012:**
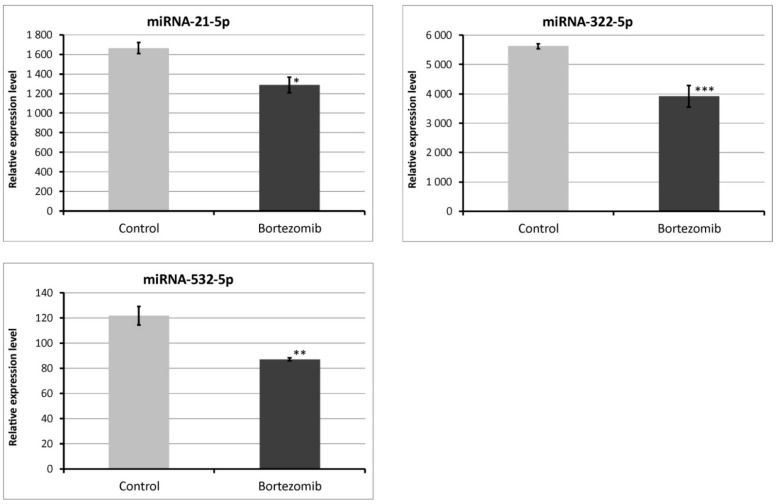
Real-time quantitation of selected miRNAs (miRNA-21-5p; miRNA322-5p; miRNA-532-5p) in PC12-derived nerve cells after bortezomib-treatment, and in control cells. Data are presented as the mean ± SD (n = 3). **p* < 0.05; ** *p* < 0.01; *** *p* < 0.002 compared with the control cells.

**Figure 13 ijms-21-00751-f013:**
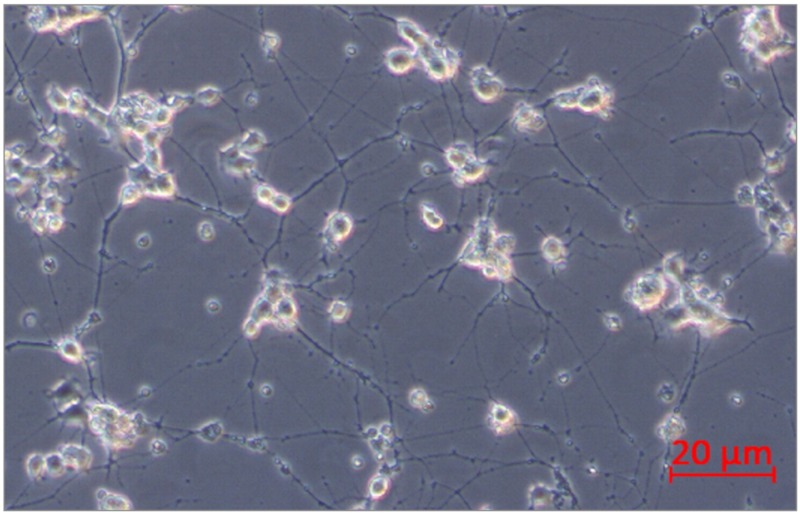
PC12 cells after 10 days of incubation with nerve growth factor (NGF).

**Table 1 ijms-21-00751-t001:** The list of 15 the most downregulated genes in bortezomib-treated PC12-derived nerve cells compared to controls.

Gene Symbol	Gene Name	Gene Function	Fold Change
*Lpl*	lipoprotein lipase	Recruited to its site of action on the luminal surface of vascular endothelium by binding to GPIHBP1 and cell surface heparan sulfate proteoglycans	−42.493
*Ddc*	dopa decarboxylase (aromatic L-amino acid decarboxylase)	important in the brain and nervous system. This enzyme takes part in the pathway that produces dopamine and serotonin, which are chemical messengers that transmit signals between nerve cells (neurotransmitters).	−38.562
*Lgr5*	leucine rich repeat containing G protein coupled receptor 5	expressed across a diverse range of tissue such as in the muscle, placenta, spinal cord and brain and particularly as a biomarker of adult stem cells in certain tissues	−32.995
*Gpr22*	G protein-coupled receptor 22	play an essential role in the regulation of cardiovascular function	−30.754
*Top2a*	topoisomerase (DNA) II alpha	controls and alters the topologic states of DNA during transcription; involved in processes such as chromosome condensation, chromatid separation, and the relief of torsional stress that occurs during DNA transcription and replication	−29.831
*Nusap1*	nucleolar and spindle associated protein 1	binds and stabilizes microtubules (by similarity); can promote the organization of mitotic spindle microtubules around them	−29.200
*Ascl1*	achaete-scute family bHLH transcription factor 1	neuronal commitment and differentiation and in the generation of olfactory and autonomic neurons	−28.366
*Hjurp*	Holliday junction recognition protein	plays a central role in the incorporation and maintenance of histone H3-like variant CENPA at centromeres	−27.183
*Mx2*	myxovirus (influenza virus) resistance 2	exhibits antiviral activity; play a role in regulating nucleocytoplasmic transport and cell-cycle progression	−24.350
*RGD1564463*	similar to Mdes protein	able to upregulation of proliferation-related gene expression, such as that of CDX2 and PCNA	−22.386
*S100a5*	S100 calcium binding protein A5	involved in the regulation of a number of cellular processes such as cell cycle progression and differentiation.	−22.222
*Bub1*	BUB1 mitotic checkpoint serine/threonine kinase	establishment of the mitotic spindle checkpoint and chromosome congression	−21.434
*Ndufa4l2*	NADH dehydrogenase (ubiquinone) 1 alpha subcomplex, 4-like 2	codes for a subunit of Complex I of the respiratory chain, which transfers electrons from NADH to ubiquinone	−20.830
*Nuf2*	NUF2, NDC80 kinetochore complex component	chromosome segregation	−19.673
*Faxdc2*	fatty acid hydroxylase domain containing 2	catalyzes the synthesis of 2-hydroxysphingolipids	−17.710

**Table 2 ijms-21-00751-t002:** The list of 15 the most upregulated genes in bortezomib-treated PC12-derived nerve cells compared to controls.

Gene Symbol	Gene Name	Gene Function	Fold Change
*Hamp*	hepcidin antimicrobial peptide	maintaining iron balance in the body	78.308
*Ankrd1*	ankyrin repeat domain 1	transcription factor involved in development and under conditions of stress	36.103
*Flrt3*	fibronectin leucine rich transmembrane protein 3	cell adhesion and receptor signalling	33.410
*Fam171b*	family with sequence similarity 171, member B	associated with neurodevelopmental disorders	22.564
*Slc7a11*	solute carrier family 7	regulates synaptic activity by stimulating extrasynaptic receptors and performs nonvesicular glutamate release	17.940
*Fam129a*	family with sequence similarity 129, member A	marker for certain cancers; regulates p53-mediated apoptosis	16.321
*Gata4*	GATA binding protein 4	transcriptional regulator for many cardiac genes; regulates hypertrophic growth of the heart; promotes cardiac morphogenesis, cardiomyocytes survival, and maintains cardiac function in the adult heart.	15.599
*Sema6a*	sema domain, transmembrane domain ™	promotes reorganization of the actin cytoskeleton;plays an important role in axon guidance in the developing central nervous system	15.091
*Myo10*	myosin X	regulates cell shape, cell spreading and cell adhesion	15.012
*Lifr*	leukemia inhibitory factor receptor alpha	controls several cellular processes, including growth and division (proliferation), maturation (differentiation), and survival	14.617
*Prss46*	protease, serine, 46	serine-type endopeptidase activity; involved in proteolysis	13.349
*Plaur*	plasminogen activator, urokinase receptor	influences many normal and pathological processes related to cell-surface plasminogen activation and localized degradation of the extracellular matrix	13.144
*Dcx*	doublecortin	involved in the movement of nerve cells to their proper locations in the developing brain, a process called neuronal migration	13.099
*Fosl1*	fos-like antigen 1	regulates tumor cell proliferation and survival	12.403
*Ddit3*	DNA-damage inducible transcript 3	adipogenesis, erythropoiesis, growth arrest and endoplasmic reticulum stress response	7.306

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
