# Peer review of "Effect of Bortezomib on Global Gene Expression in PC12-Derived Nerve Cells"

_ijms, 2020, doi:10.3390/ijms21030751_

Round 1
Reviewer 1 Report
Łuczkowska et al. studied the effects of Bortezomib on neural cells. In particular, they focused the effects through miRNAs and mRNA expression levels. As the authors indicate, the study is rather preliminary, with weakness such as the use of just one dose of the drug at one time point. However, it provides data that might be interesting for future work. There are some minor issues that should be addressed: Introduction: Page 2, lines 61-63: Eliminate the repeated sentence. Fold changes, False Discovery Rate (FDR), relative expression levels in Fig 9, material and methods (for example, line 285, 344,…), etc. have a comma instead of decimal point. Figure 2 and 3, it seems redundant to use a colour code (red and blue) and two columns. If it is in red, meaning upregulated in controls, it must be downregulated in bortezomib-treated cells. At least, this should be better explained. Since PC12 is a rat cell line, genes should be written italicized, with only the first letter in upper-case. In the text, there is a mixture of genes written italicized or not, with just the first letter in upper-case or all the letters in upper-case. Thus, it is not clear if the authors refer to human or rat genes. Reference Lee et al., line 209, has no number and is not in the reference list. Reference 13 (line 450) eliminate the change of line between “multiple” and “myeloma”. Probably “10.26402” should be added after “60(2)”. Reference 29 is incomplete. Add ”:45101”.
Reviewer 2 Report
This study aimed to investigate the putative molecular mechanisms underlying the development and progression of bortezomib-induced peripheral neuropathy.
Comments
It is well known that peripheral sensory nerves - under normal conditions at least - don’t replicate and don’t undergo mitotic cell division in contrast to PC12 cells which do. Therefore the main finding (page 14) “… we demonstrated for the first time the global effect of bortezomib on expression of genes and miRNAs associated with neurogenesis, cell cycle, cell division, DNA repair in nerve cells” is correct for PC12 cells but not for dorsal root ganglion neurons which differ quite remarkably. Moreover, in clinical practice is the repeated administration of bortezomib and not only a single exposure the cause for the development of a neuropathy. Therefore, there is no justification to write about possible mechanisms for “bortezomib-induced peripheral neuropathy”. This should be omitted throughout the entire manuscript. What the manuscript shows is the transcriptional effects of a single exposure of bortezomib on undifferentiated PC12 cells.
Round 2
Reviewer 2 Report
The authors have addressed all comments sufficiently.